# Evaluation of Posturographic and Neuromuscular Parameters during Upright Stance and Hand Standing: A Pilot Study

**DOI:** 10.3390/jfmk8020040

**Published:** 2023-03-30

**Authors:** Ewan Thomas, Carlo Rossi, Luca Petrigna, Giuseppe Messina, Marianna Bellafiore, Fatma Neşe Şahin, Patrizia Proia, Antonio Palma, Antonino Bianco

**Affiliations:** 1Sport and Exercise Sciences Research Unit, Department of Psychology, Educational Science and Human Movement, University of Palermo, Via Giovanni Pascoli 6, 90144 Palermo, Italy; ewan.thomas@unipa.it (E.T.); giuseppe.messina17@unipa.it (G.M.); marianna.bellafiore@unipa.it (M.B.); patrizia.proia@unipa.it (P.P.); antonio.palma@unipa.it (A.P.); antonino.bianco@unipa.it (A.B.); 2Department of Biomedical and Biotechnological Sciences, Section of Anatomy, Histology and Movement Science, School of Medicine, University of Catania, Via S. Sofia n°97, 95123 Catania, Italy; luca.petrigna@unict.it; 3Department of Coaching Education, Faculty of Sport Science, Ankara University, Ankara 06830, Türkiye; nese.sahin@ankara.edu.tr

**Keywords:** handstand, postural control, postural balance, sEMG, stabilometric assessment, exercise

## Abstract

Upright bipedal posture is the physiological human posture; however, it is not the only possible form of human standing; indeed, an inverted position, a handstand, is required during gymnastics or other sports. Thus, this study aimed to understand the differences between the two standing strategies from a postural and neuromuscular perspective. Thirteen gymnasts with at least three years of sports experience underwent a baropodometric assessment and a surface electromyography (sEMG) examination in a standard upright bipodalic stance and during a handstand. The sEMG examination was performed on the gastrocnemius during an upright stance and on the flexor carpi radialis during the handstand. Limb weight distribution presented differences between the two vertical stances (*p* < 0.01). During the handstand, the weight ratio was prevalently observed on the palm of the hand for both hands with a significant difference between the front and rear aspect of the hand compared to the standing tasks (*p* < 0.01). Normalized sEMG amplitude showed significant differences during bipedal standing and hand standing; however, over a 5 s period, the normalized median frequency (MDF) value was similar for the two tasks. Both standing tasks presented similar postural weight managing patterns when analysed on the frontal plane, but they were different on the sagittal plane. In addition, the neuromuscular patterns during a 5 s window differ in amplitude but not for the frequency domain.

## 1. Introduction

Postural control is the ability of the nervous system to modulate the distribution and geometry of mass, as well as position and orientation through the integration of information from the vestibular, visual, and tactile apparatus [1]. Humans develop balance control strategies in postural positions at an early age by arranging muscle contractions to create torques around joints to preserve the center of mass (CoM) projection within the region of stability at the base of the support [2,3]. However, postural perturbation experiments have shown that ankle, hip, ankle-hip, and ankle-knee-hip strategies can be rapidly organized into a reactive mode to preserve balance stability [4,5]. It must be considered that bipedal postural control is not the only posture; in fact, gymnasts and people who work in the circus also adopt other forms of standing positions, such as hand standing [6]. As with the bipedal posture, the hand standing also integrates information from vision, vestibules, and tactile resolution of the hands [1,7]. The handstand is a fundamental skill in gymnastics in both men’s and women’s and is also included in more complex gymnastic skills [8]. The limited stability region of the handstand leads to a requirement for a balance control strategy that is more constrained than that for standing [1,9,10]. In addition, the handstand motion is a less natural motion, which can cause a feeling of unsteadiness or disorientation which results in the need for a more constrained balance control strategy than standing [4]. Thus, the main difference between the two types of postures could be due to the different muscles and joints adopted and, consequently, to the different strategies that the body has to adopt to maintain such posture. The hand standing posture, like the bipedal standing posture, also adopts three joint levels [1]. During handstands, it has been noted that the body begins to move earlier in the distal body segments while the trunk and head remain more stable [10]. Similarly, during the bipedal position, the ankle is the first joint which gets involved, while, during the handstand, the activation of the wrist flexors is the primary strategy to maintain balance [11]. This could be explained by the closer position of the wrist to the contact point on the ground and the need to minimize upper body movement [12]. Apparently, the best balances during the handstand are represented by major contributions from wrist and shoulder torques with little influence from hip torques. On the contrary, there is increasing awareness regarding the importance of hip torques, which are more influential in less successful balances [7,12]. In regards to the bipedal posture [13], the experience carried out with gymnastics athletes, young people, and adults have demonstrated a different relationship between muscle activity and postural control variables; furthermore, there is a reduction of the pressure shift and, consequently, less muscle activity related to the use of the wrist flexors together with the anterior deltoid [11].

The two tasks share common processes such as the sense and movement of the body and the implementation of a multi-joint strategies to preserve the orientation within certain constraints. Consequently, understanding mechanisms or strategies in one task can lead to insights into the other [7]. Additionally, people use a small set of preferred compensatory movement strategies to preserve balance during the handstand [10].

One study using two support surfaces, one foam and the other solid, highlighted the difficulty that even experienced people have in controlling performance during hand standing on non-solid surfaces [14]. Furthermore, signal delays are an important feature of any biological system, with large delays reducing stability and complicating control, and, consequently, this will be evaluated through surface electromyography (sEMG). Handstand balance performed by experienced gymnasts provides an alternative perspective to a normal standing position for understanding this complex system. Therefore, the present study aimed to compare the bipedal position and the hand standing position by evaluating the two tasks from a baropodometric and neuromuscular point of view in gymnasts. In this study, we wanted to understand the differences between the upright bipedal and vertical posture, in terms of body weight distribution and neuromuscular activation. We assumed that there would be differences in body weight distribution and neuromuscular patterns between the two postures.

## 2. Material and Methods

### 2.1. Participants

A total of 13 gymnasts with at least three years of sports experience, 10 males and 3 females (mean and standard deviation: age (years) 20.2 ± 4; height (cm) 170.4 ± 7.4; weight (kg) 66.0 ± 11.0), were retained for investigation. Participants were included if they were free of injuries during the assessment period and if they had an experience of at least three years in training with a handstand. Each gymnast regularly exercised ~1 h daily from Monday to Friday. Before the study, participants were informed about the study protocol, and the risks and benefits of their participation, and they gave written individual informed consent to participate in the study. The study was approved by the local Bioethics Committee of the University of Palermo (ref. n°121/2023) and was carried out according to the principles established by the Declaration of Helsinki.

### 2.2. Study Design

The total duration of each assessment lasted around 30 min, during which participants compiled and signed the documentation. Subsequently, anthropometric measurements were taken. Once all individuals’ data were collected, each participant had to undergo a maximal voluntary contraction (MVC). Following the MVC, a sEMG evaluation was performed for the gastrocnemius and the flexor carpi radialis together with a baropodometric assessment. The documents signed by the participants were related to their consent to take part in the study and their permission to use their data. Anthropometric measures were related to height, evaluated with a meter, and weight, measured with a professional balance Seca scale (maximum weight recordable: 300 kg; resolution: 100 g; Seca, Hamburg, Germany). The baropodometric assessment was performed in an upright stance and during a handstand (Figure 1). For each task, the sEMG examination of the gastrocnemius during an upright stance and of the flexor carpi radialis during the handstand was retained for analysis. Before the assessment, participants performed a light warm-up consisting of 1 set of dynamic stretching of the targeted muscles of 30 s duration [15].

### 2.3. Baropodometric Assessment

All tasks were performed on a FreeMed baropodometric platform (50 × 60 cm) and with the FreeStep v.1.0.3 software. The sensors, coated with 24 K gold, guaranteed the repeatability and reliability of the instrument (produced by Sensor Medica, Guidonia Montecelio, Roma, Italy). During the upright stance, the gymnasts had to stand with feet parallel, and they had to keep that position for the duration of the assessment. The upper limbs were completely extended. Participants had to fixate on a point 2 m ahead, on a white surface, at eye level. The length of the trial was 5 s. No indications were given by the investigator related to the distance between the feet or the position to keep during the assessment not to influence the posture of each participant. However, reference points in the platform assessment area had to be maintained. Participants were barefoot.

During the handstand, the hand position was chosen by the participants. As above, reference points in the platform assessment area had to be maintained. The length of the trial was 5 s. Changes in support surface or the contact of the floor with the lower limbs was considered a failure of the task. If the participant failed the task, this could be repeated a maximum of 3 times after a 5 min break to avoid muscle fatigue.

Variables retained from the baropodometric assessment concerned: Support surface for each limb (cm^2^) with information regarding the total support surface and the front and rear aspect of each analysed limb (based on the acquired image with a 50% ratio between front and rear aspect). Further, weight distribution (%) for the same variables mentioned above were also retained.

### 2.4. sEMG Evaluation

Before the sEMG evaluation, an MVC was performed. The maximal isometric force exerted by the gastrocnemius muscle and the flexor carpi radialis muscle were determined by asking the participants to increase the force from rest to maximum gradually in ∼3 s and to then maintain the maximum for an additional 3 s. Repeated contractions were performed until 2 attempts were within 5% of each other, and the greater peak force was used as the subject’s MVC force [16]. A customized apparatus was built to perform the MVC of the gastrocnemius muscle and the flexor carpi radialis muscle.

Before the sEMG evaluation, participants’ skin was shaved (if necessary) and cleaned with alcohol at 70%. The signals were collected using an OTBio Quattro EMG device (Copyright@ 2010 OT Bioelettronica, Torino, Italy), unilaterally, from the dominant side of the body using a bipolar method. The surface electrodes (Ag/AgCl sEMG disposable adhesive circular bipolar surface electrodes of 24 mm diameter, CDE02401500BX, Spes Medica S.r.L, Battipaglia, Italy), were placed with a distance of 1 cm between them on the bellies of the gastrocnemius and the flexor carpi radialis muscles. Electrode location was determined according to the recommendations provided by Barbero et al. [17] and the manufacturer’s instructions.

The reference electrode (20 × 25 mm snap disposable electrodes, DENIS, Spes Medica S.r.L, Battipaglia, Italy) was placed on the malleolus of the fibula during the evaluation of the gastrocnemius muscle and on the styloid process of the radius during the evaluation of the flexor carpi radialis muscle (Figure 1).

All sEMG data was bandpass filtered using cut-off frequencies between 20 and 400 Hz. The signal was then processed to obtain the amplitudes root mean square (RMS) and the median frequency (MDF) for either the standing or hand-standing task. The obtained values were then normalized according to each participant’s MVC for each task.

### 2.5. Statistical Analysis

Means, standard deviations, and 95% confidence intervals (CI) were calculated to present data. Inferential statistics were carried out with Jamovi (The jamovi project (2021). jamovi (Version 1.8.0.1) [Computer Software]. Retrieved from https://www.jamovi.org). A Shapiro—Wilks test was performed to identify the normality of the distribution of all parameters. Parametric and non-parametric assessment was adopted when appropriate. To assess the differences in stabilometry and sEMG between tasks for each participant a paired *t*-test or a Wilcoxon test were adopted when appropriate. For each analysis, the effect size (ES) was calculated. Cohen δ’s were adopted as measures of ES for parametric data, while biserial rank correlation was used for non-parametric data. The magnitude of the Cohen δ’s ES was classified according to the following scale: 0–0.19 = trivial effect, 0.20–0.49 = small effect, 0.50–0.79 = moderate effect, and  ≥0.80 = large effect [18], while the magnitude for the biserial rank correlation were classified as 0–0.1 = trivial effect, 0.10–0.3 = small effect, 0.30–0.5 = moderate effect, and  ≥ 0.50 = large effect [19]. Graphs were created with GraphPad Prism8 (GraphPad Software, San Diego, CA, USA). The significance for all analyses was set at *p* < 0.05.

## 3. Results

Weight is equally distributed during the upright stance with no differences between the left and the right limb (50.3 ± 1.8% and 49.7 ± 1.8%, respectively). Similarly, also during hand standing, no statistically significant differences were detected in the weight distribution between left and right limbs (45.1 ± 4.9% and 54.9 ± 4.9%, respectively) (Figure 2). However, there were significant differences between the two tasks (upright stance vs. handstand) (Table 1).

Regarding the analysis of the base of support during the upright stance, the forefoot and rear part of the foot were similar (F/R 49/51% weight distribution).

Conversely, during the hand-standing task, the ratio was prevalently on the palm of the hand (F/R 27/73% weight distribution). This difference was significant, and it was noted for both hands (*p* < 0.01).

Normalized sEMG amplitude (RMS) showed a significant difference (*p* < 0.01) during bipedal standing and hand standing (5.9 ± 2.6 vs. 92.9 ± 45.8% activation) (Figure 3). However, over a 5 s period, the normalized MDF value (Figure 4) was similar for the two tasks (71.2 ± 18.5 vs. 71.9 ± 14.0, standing vs. hand standing, respectively) (Table 2).

## 4. Discussion

The purpose of the following study was to understand the differences between the standing strategy and the hand standing from a postural and neuromuscular perspective and to evaluate if any differences were present. The main results of this study highlight similarities between the two tasks regarding weight distributions between the two limbs, with differences within each limb across tasks. Concerning the normalized sEMG amplitude, there were significant differences during the bipedal position and the hand standing. The handstand is considered a fundamental gymnastic skill [8]. On the other hand, the variability of acceleration during the inverted positions could serve as a source of information to preserve stability [10]. For weight management strategies in the sagittal plane, we found a difference between the two tasks. Furthermore, the neuromuscular patterns during a 5-s window differ in the time-intensity domain but not in the frequency domain. The ability to orient body parts concerning gravity, support surface, visual environment, and internal references is a critical component of the postural control [2]. A healthy nervous system automatically changes the way the body is oriented in space, depending on the context and the task [2].

The level of experience of the sportsman considerably differentiates the stability of an upright body. In fact, subjects with greater experience who practice gymnastics, compared to young people, are characterized by a better ability to control the position of the body in both positions [20]. Hand standing has been seen to provide a larger center of pressure (CoP) area than the upright postural position, which could be caused by a smaller contact area which reduces the base of support of the body, and a reduction in postural control due to the unusual balancing task [10,21]. In addition, hand stand experience has been associated with better performances of postural tasks [14,22]. Based on the literature on bipedal initiation, postural adjustments that help achieve optimal balance and propulsive requirements for gait execution are performed in the movement preparation phase [23,24,25,26,27]. Grabowiecki et al., 2021 proposed that the anterior displacement strategy is implemented when the CoP is closest to the posterior boundary of the BoS (base of support) and behind the vertical projection of the CoM at the onset of walking initiation during handstand [28]. Conversely, the posterior displacement strategy is performed when the CoP position is beyond the vertical projection of the CoM and closer to the anterior border of the BoS.

As a result, the reduced surface area of the hand acts like the foot to maintain body balance with the wrists and shoulders acting as the ankle and hip (strategy adopted during bipedal stance) [11]. It is well-known that keeping a straight body shape without any angles in the shoulder, elbow, hip, and knee joints and a strong balance between agonists and antagonist’s muscles is required for high-quality handstand postures [29,30,31]. Therefore, such findings highlight that shoulder torque plays a more important role than hip torque in the handstand postural control strategy [7,12,32]. Concerning “ankle strategy” and “hip strategy” in standing [33], the hip specificities of gymnasts seem relatively uncoordinated and arbitrary at least until the wrists and shoulders work mainly to regulate postural balance [22]. Therefore, future research on this topic should consider a study on the ‘‘wrist strategy’’.

We also found significant values for the forefoot and hindfoot, both right and left with the respective hand. In the study by Yamazaki et al., 2005 it was seen that the trajectories of the foot pressure center varied, initially towards the Rt side and then towards Lt which, respectively, coincided with the initial and subsequent phases of the trunk rotations and muscle activation [34]. Postural adjustment in the handstand would appear to be organized according to a system similar to that of upright posture, with joint levels suggesting the existence of an organization typical of human posture [35]. The results of this study can provide important insights into human motor performance in different positions and how postural and neuromuscular factors interact with each other. This can be especially useful for athletes and professionals who require exceptional postural stability and superior neuromuscular strength.

Regarding the sEMG amplitude, we observed a significant difference between the standing and hand standing tasks. It was expected that such differences were present, since hand standing is a complex skill. The results obtained by the present investigation concerning the high activation of the wrist flexors are similar to those of Kochanowicz et al. [1]. The wrist flexors in the context of hand standing act like the plantar flexors during conventional standing [1]. On the other hand, the sEMG amplitude during quiet standing showed very low values, which are in line with existing literature for resting muscle activity during balancing tasks [36]. In the frequency domain, we found no significant differences between the hand standing and the standing tasks, unlike the study by Wyatt et al. 2021, which, instead, showed through a multiple regression that the frequency domain is an excellent predictor of the duration of the handstand equilibrium [37]. This is a metric that is usually used to identify patterns of fatigue. Probably, the five-second window adopted in the present study is too short to cause fatigue [38]. Secondly, the sample was composed of expert gymnasts which are used to performing the handstand task, therefore preventing any variation in the frequency discharge of the involved muscles [11,20]. Recruitment strategies at the time of measurement were similar for upper and lower limbs during static postural stability tasks [39].

Their results showed that the alternating rotations of the upper torso, produced by rapid arm movements, were transmitted to the hip in part due to co-contraction of the trunk muscles, and each pair of muscles in the hip joint contributed to maintaining upright posture by stabilizing the hip joints against alternating rotations of the trunk [35]. The percentage of time spent in different control strategies for perturbed and unperturbed standing and handstand balance was determined [4], during both perturbed and unperturbed balance, the predominant control strategies were a standing ankle strategy and a vertical wrist strategy. Findings reveal that the central nervous system maintains balance during a variety of tasks and postures by employing an individual control strategy [4].

Limitations of the present study are the small number of recruited participants and also the specific population analyzed concerning only young gymnasts. Another limit of the study was that no control was performed on athletes’ previous injuries. Future studies should also include a greater proportion of women, other age groups, and people with different sporting backgrounds who can perform a handstand.

## 5. Conclusions

Postural and neuromuscular differences were observed between standing and hand standing. During a handstand, we observed that the majority of the support surface was attributed to the area of the palm of the hand, while an even distribution between hind foot and rear foot was observed during the standing task. Furthermore, sEMG muscle activity was different between the two positions with greater effort during hand standing vs. standing. However, analysis of frequency was similar across the two positions over a 5 s time frame. More research is needed to gain insight into the different forms of standing postures.

## Figures and Tables

**Figure 1 jfmk-08-00040-f001:**
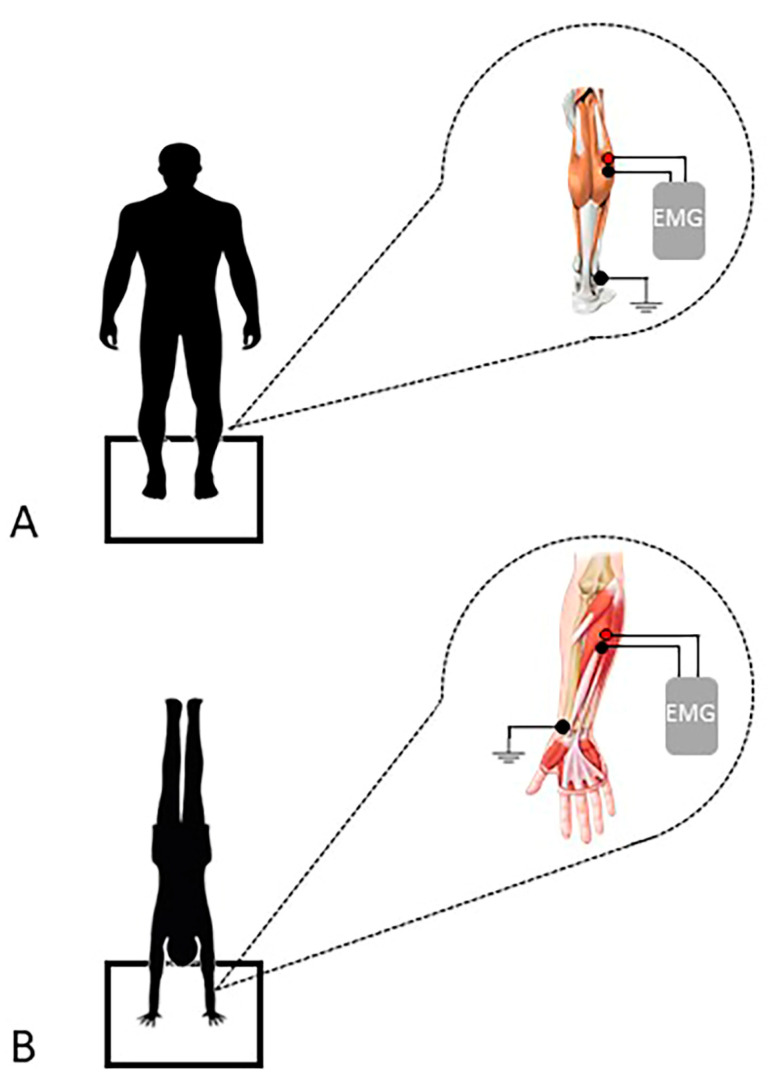
Graphical depiction of the experimental setting. In panel (**A**), the baropodometric evaluation and the sEMG applied to the gastrocnemius muscle during the standing task. In panel (**B**), the baropodometric evaluation and the sEMG applied to the flexor carpi radialis muscle during the hand standing task.

**Figure 2 jfmk-08-00040-f002:**
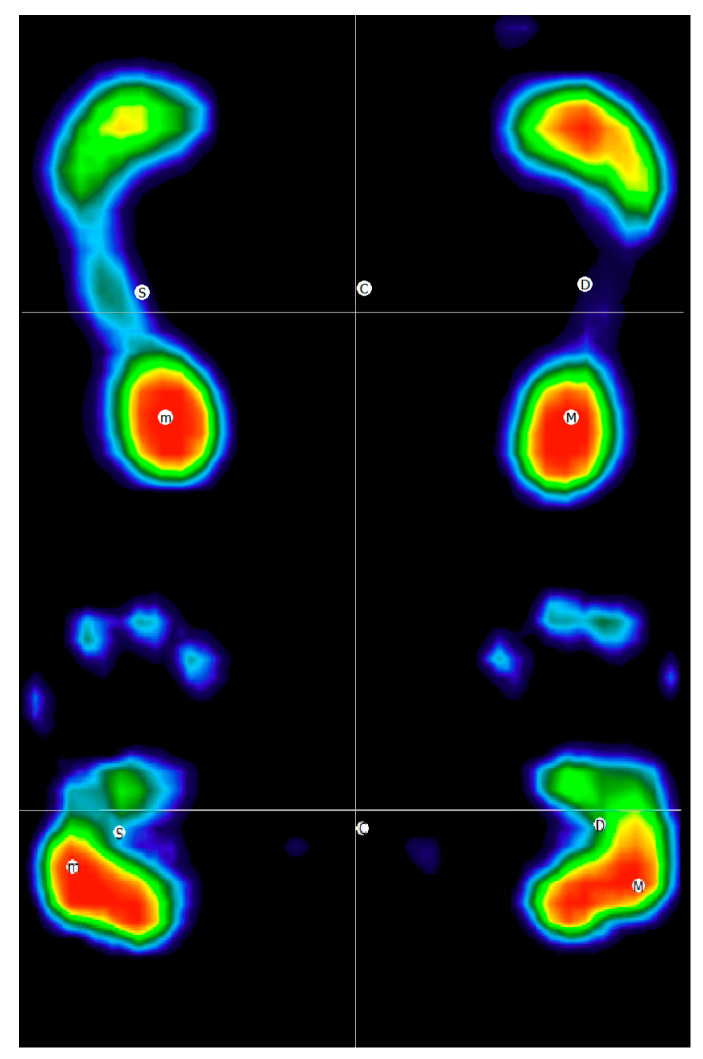
Representation of the standing and hand standing task during the baropodometric evaluation of one participant.

**Figure 3 jfmk-08-00040-f003:**
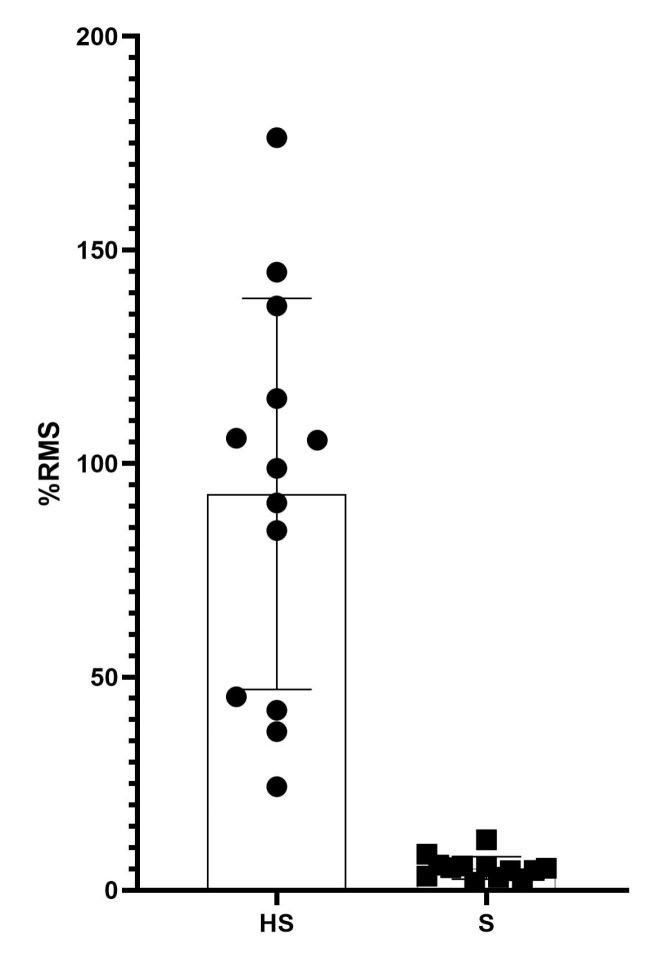
Figure represents the mean and studv of sEMG amplitudes RMS (root mean square) percentages of the two performed tasks of the analyzed sample HS = Handstand; S = Standing.

**Figure 4 jfmk-08-00040-f004:**
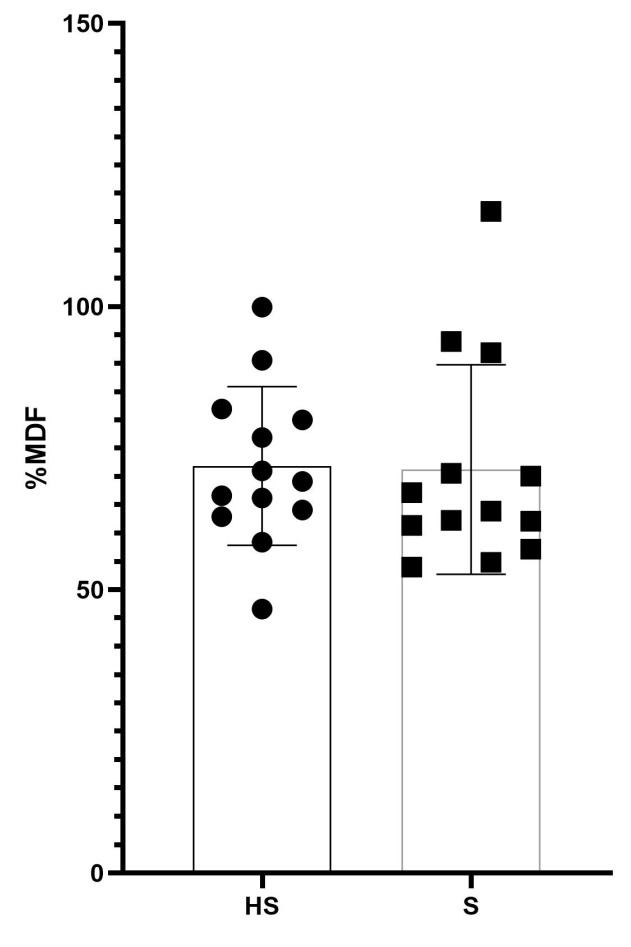
Figure represents the mean and study of sEMG MDF (median frequency) percentages of the two performed tasks of the analyzed sample. HS = Handstand; S = Standing.

**Table 1 jfmk-08-00040-t001:** Descriptive measures of the baropodometic assessment during standing and hand standing.

	Standing	Hand Standing	*p*	ES	CI (95%)-S	CI (95%)-HS
Left supporting surface (cm^2^)	126.1 ± 22.7	86.6 ± 19.2	<0.01 §	1.75	112.5–139.8	75–98.2
Right supporting surface (cm^2^)	122.1 ± 21.2	90.8 ± 15.2	<0.01 §	2.69	109.4–135	81.6–99.9
Left forefoot/hand (cm^2^)	70.4 ± 13.8	32.9 ± 7.8	<0.01 §	2.70	62.1–78.7	28.2–37.7
Right forefoot/hand (cm^2^)	67.8 ± 13.9	34.3 ± 8.1	<0.01 §	2.30	59.5–76.2	29.4–39.2
Left backfoot/hand (cm^2^)	55.7 ± 10.1	53.7 ± 12.9	0.60 §	0.14	49.6–61.8	45.9–61.5
Right backfoot/hand (cm^2^)	54.4 ± 9.3	56.4 ± 8.3	0.45 §	0.21	48.8–60	51.4–61.4
Left supporting surface (%)	50.3 ± 1.8	45.1 ± 4.9	<0.01 #	0.91	49.2–51.4	42.2–48.1
Right supporting surface (%)	49.7 ± 1.8	54.9 ± 4.9	<0.01 #	0.91	48.6–50.8	51.9–57.8
Left forefoot/hand (%)	49.1 ± 7.3	27.6 ± 6.1	<0.01 §	2.17	44.8–53.5	24.3–30.9
Right forefoot/hand (%)	47.8 ± 6.6	25.1 ± 5.4	<0.01 §	2.30	43.8–51.7	21.5–28.8
Left backfoot/hand (%)	50.8 ± 7.3	72.4 ± 6.1	<0.01 §	2.17	46.5–55.2	69.1–75.7
Right backfoot/hand (%)	52.2 ± 6.6	74.9 ± 5.4	<0.01 §	2.30	48.3–56.2	71.2–78.5

§-parametric evaluation; #-non-parametric evaluation; ES-effect size; CI-confidence interval; S-standing; HS-hand standing.

**Table 2 jfmk-08-00040-t002:** sEMG Measures of both standing and hand standing.

	Standing	Hand Standing	*p*	ES	CI (95%)-S	CI (95%)-HS
MVC (μV)	194.6 ± 147.9	405.6 ± 200.6	0.01 *#*	0.78	105–284	284–527
Mean Amplitude (μV)	11.5 ± 12.6	326.1 ± 145.3	<0.01 *#*	1.00	3.9–19.2	238–414
MDF (Hz)	164.4 ± 30.0	99.3 ± 28.3	<0.01 §	1.54	146–183	82.2–116
Mean Frequency (Hz)	114.4 ± 23.2	69.1 ± 15.1	<0.01 §	2.13	100.3–128.4	60–78
Normalized Amplitude (%RMS)	5.2 ± 2.6	92.9 ± 45.8	<0.01 §	1.90	3.6–6.8	65.2–120.6
Normalized Frequency (%MDF)	71.2 ± 18.5	71.9 ± 14.0	0.54 *#*	0.21	60–82.4	63.4–80.3

§-parametric evaluation; #-non-parametric evaluation; ES-effect size; CI-confidence interval; S-standing; HS-hand standing.

## Data Availability

Data are available from the corresponding author upon reasonable request.

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
