# Peer review of "Evaluation of Posturographic and Neuromuscular Parameters during Upright Stance and Hand Standing: A Pilot Study"

_jfmk, 2023, doi:10.3390/jfmk8020040_

Round 1
Reviewer 1 Report
Main impression
This study examines an interesting question that can provide fundamental insights into the neuromuscular control of posture. The topic of the manuscript attracted my attention and I was eager to read the findings. Unfortunately, the manuscript fell short of my expectations and I can identify several aspects that should be thoroughly revised before I can recommend this study for publication. With respect to the authors’ work, I list below my concerns and suggestions, and I hope that I can review a revised version that provides more transparent results that can be more easily digested.
Major comments
One of my main comments is that authors should provide a clear illustration of their setup and conditions. For this, a graphical depiction should be included in the manuscript. There, the reader can understand better what exactly were the imposed tasks. At this moment, this is unclear. A setup figure should complement the following text suggestions so that the manuscript is more to the point and provides transparent information about the task.
In addition, the results should be illustrated with appropriate figures. First of all, the presentation of raw data would significantly contribute to the understanding of what exactly the variables are. The authors could include some figures with raw (or filtered) EMG time courses for each limb and each condition.
In addition, the main results such as RMS of EMG and muscle activity of each limb in each condition should be presented in figures with bar or point plots, including suitable error bars ^and^ individual data points.
Other comments
Line 50: do vestibular signals not get involved in the control of hand standing?
Line 52: Why limited region? The size of the hands is not much smaller than the size of the feet, so the stability region should be similar. Please revise or explain.
Lines 76-77: unclear whether this sentence refers to bipedal or hand standing. Please revise to clarify.
Line 113: Is it correct that participants could choose how wide apart would their feet and hands be, during the bipedal and hand standing condition, respectively? If yes, why do authors not control for that? If the base of support is wider, then the postural challenges may be compromised. The authors should first clarify this aspect and if the base of support is not identical between conditions and subjects, they should provide the information for each subject, and ideally put this measure as a covariate in their analyses.
Lines 127-128: I am surprised to read this sentence. What does it mean that there is a forefoot/hand and a backfoot/hand? Weren’t the limbs parallel to each other? If not, this should be explicitly clarified. How far apart were the limbs in the anteroposterior (and mediolateral, please see previous comment) direction? Which limb was placed back and why? Was this chosen by the participants or was it controlled by the experimenters? Related to this, did the authors obtain the handedness and footedness of each participant, and if yes, with which method? If this is obtained, please report the respective results. If not, please discuss how this factor may have influenced the results.
Table 1: Unclear what are all these variables, as well as the reported units. Please provide explicit information about what variables were calculated, and please describe these calculations in sufficient detail. The manuscript at this stage promises that there will be three variables (RMS of EMG signal, median frequency, and relative muscle contraction –even though the description of these is quite superficial). For a fair evaluation of the results, and for transparent research practices, the authors are kindly requested to explain in sufficient detail which exact variables they calculated and why, as well as how these were calculated.
Lines 167-168: Unclear what exactly is this effect. The authors refer to hands but while standing. Do they refer to feet during bipedal standing, or to hands during hand standing? Also, were the front limbs (hand or foot) taking more or less of the mass distribution? Please clarify the effect.
Line 171: What do the percentages in the brackets actually represent?
Results: Where are the results of the RMS of the EMG?
Line 195: what is a higher CoP? Do the authors mean 'more variable'? Or a CoP that has a larger peak-to-peak displacement? The current formulation is imprecise.
Minor comments
There are several typos throughout the manuscript, such as:
. line 45, ‘ankle, knee-hip’, requires a hyphen between ankle and hip,
. line 51, ‘in gymnastics’ is repeated twice in a row,
. line 114, ‘fix’ should be ‘fixate’,
. line 114, ‘high’ should be ‘height’,
. line 165, ‘for each foot’ should be removed,
, and so on, please proofread once more the manuscript and fix inconsistencies in style (such as the font size in line 167) and other typos.
Author Response
REVIEW 1
Dear reviewer, thanks for the comments provided and for the time dedicated to reviewing our manuscript.
Below, we have provided a detailed response to each comment.
Thanks again.
Major comments
- One of my main comments is that authors should provide a clear illustration of their setup and conditions. For this, a graphical depiction should be included in the manuscript. There, the reader can understand better what exactly were the imposed tasks. At this moment, this is unclear. A setup figure should complement the following text suggestions so that the manuscript is more to the point and provides transparent information about the task.
Dear reviewer, a graphical depiction has been included as suggested. Thanks for the comment.
- In addition, the results should be illustrated with appropriate figures. First of all, the presentation of raw data would significantly contribute to the understanding of what exactly the variables are. The authors could include some figures with raw (or filtered) EMG time courses for each limb and each condition.
Thank you for your comment. Figures have been added as suggested.
- In addition, the main results such as RMS of EMG and muscle activity of each limb in each condition should be presented in figures with bar or point plots, including suitable error bars ^and^ individual data points.
Thank you for your comment. The figure has been added.
- Line 50: do vestibular signals not get involved in the control of hand standing?
Thanks for your important comment. We acknowledge the oversight and have added the vestibules as an important signal in vertical control.
- Line 52: Why limited region? The size of the hands is not much smaller than the size of the feet, so the stability region should be similar. Please revise or explain.
Thanks for your comment. We have better explained the concept of the greater control required vertically.
- Lines 76-77: unclear whether this sentence refers to bipedal or hand standing. Please revise to clarify.
Thanks for your comment. We specified what the task was.
- Line 113: Is it correct that participants could choose how wide apart would their feet and hands be, during the bipedal and hand standing condition, respectively? If yes, why do authors not control for that? If the base of support is wider, then the postural challenges may be compromised. The authors should first clarify this aspect and if the base of support is not identical between conditions and subjects, they should provide the information for each subject, and ideally put this measure as a covariate in their analyses.
Dear reviewer, there is a large debate regarding what should be the correct assessment procedure on baropodometric platforms. We are aware that changing the distance between the feet and/or hands may change the postural measured outcome. One thing to be considered, however, is that the platform is 50 x 60 cm. While seemingly this could represent a “large surface”, the internal surface which acquires the signal is relatively smaller. With this said, each participant was free to position his feed or hands, according to his/her natural posture (internal or external foot/hand rotation), while the stance for both feet and hands falls within the borders of the measurement surface. Therefore, no differences in the base of support of the participants could be performed. We hope that the graphical depiction can also clarify such aspect. Thanks again.
- Lines 127-128: I am surprised to read this sentence. What does it mean that there is a forefoot/hand and a backfoot/hand? Weren’t the limbs parallel to each other? If not, this should be explicitly clarified. How far apart were the limbs in the anteroposterior (and mediolateral, please see previous comment) direction? Which limb was placed back and why? Was this chosen by the participants or was it controlled by the experimenters? Related to this, did the authors obtain the handedness and footedness of each participant, and if yes, with which method? If this is obtained, please report the respective results. If not, please discuss how this factor may have influenced the results.
Dear reviewer, I believe there has been a misunderstanding. The baropodometric platform measures weight distribution and surface for each limb. According to reference values provided automatically by the manufacturer, the software provides information for each analyzed limb concerning its distribution on the anterior or posterior aspect. Therefore, we confirm that the limbs for each of the two tasks were positioned parallel to each other. These parameters are provided because even if each limb may provide a similar weight pattern, the weight management strategy within each limb may differ.
- Table 1: Unclear what are all these variables, as well as the reported units. Please provide explicit information about what variables were calculated, and please describe these calculations in sufficient detail. The manuscript at this stage promises that there will be three variables (RMS of EMG signal, median frequency, and relative muscle contraction –even though the description of these is quite superficial). For a fair evaluation of the results, and for transparent research practices, the authors are kindly requested to explain in sufficient detail which exact variables they calculated and why, as well as how these were calculated.
Dear reviewer, the manuscript aimed to identify both Posturographic and neuromuscular data. Table 1 reports Posturographic data. While table 2 reports the neuromuscular data. Concerning the presentation of the included variables, these have been further described within the method section. Thanks for the comment.
- Lines 167-168: Unclear what exactly is this effect. The authors refer to hands but while standing. Do they refer to feet during bipedal standing, or to hands during hand standing? Also, were the front limbs (hand or foot) taking more or less of the mass distribution? Please clarify the effect.
Dear reviewer after careful reading we realized it was an error in the presentation of the data. Thanks for recognizing it. We modified the sentence accordingly. Thanks again.
- Line 171: What do the percentages in the brackets actually represent?
The percentages are relative to normalized frequency. A sentence has been added for clarity.
- Results: Where are the results of the RMS of the EMG?
Dear reviewer, the results of the RMS of the sEMG are presented in Table 2 . We have modified the table accordingly, thanks.
- Line 195: what is a higher CoP? Do the authors mean 'more variable'? Or a CoP that has a larger peak-to-peak displacement? The current formulation is imprecise.
Dear reviewer, we have modified the sentence for clarity. Thanks.
- Minor comments
- There are several typos throughout the manuscript, such as: line 45, ‘ankle, knee-hip’, requires a hyphen between ankle and hip, line 51, ‘in gymnastics’ is repeated twice in a row, line 114, ‘fix’ should be ‘fixate’, line 114, ‘high’ should be ‘height’, line 165, ‘for each foot’ should be removed, , and so on, please proofread once more the manuscript and fix inconsistencies in style (such as the font size in line 167) and other typos.
Thanks for your valuable comments. We have fixed the errors.
Reviewer 2 Report
The present study aims to compare the bipedal position and the hand standing position by evaluating the two tasks from a stabilometric and muscular point of view in gymnasts.
The results highlighted that the differences were observed between the upright bipedal position and the hand standing position.
I have several suggestions.
ABSTRACT:
- “Thus, this study aimed to understand the differences between the two standing strategies from a postural and neuromuscular perspective” - it will be better to put here the purpose of the work, which is included in the introduction section: to compare the bipedal position and the hand standing position by evaluating the two tasks from a stabilometric and muscular point of view in gymnasts.
- Please expand the abbreviations used in the abstract section (EMG, MDF)
MATERIAL AND METHODS
- “Participants were included if they were free of injuries during the assessment period” - injuries they suffered earlier - e.g., a month before the study could have affected the results. Has this been considered? If not, it should be in the limitation section.
- “The study was approved by the local Bioethics Committee” - please provide the name of the organization issuing the approval (what university, etc.)
- “The surface electrodes, according to manufacturer instructions, were placed at a distance of 1 cm between them on the bellies of the gastrocnemius and the flexor carpi radialis muscles” - Was any electrode placement system used? (e.g., SENIAM standards)
- Statistical analysis - please provide a program for statistical analysis, was the sample size calculated? Describe the tests used in detail.
OVERALL:
Please improve the style, language, and interpunctions in the paper. In general, the work is interesting and can contribute to the literature. I hope my suggestions will help improve this work.
Author Response
REVIEW 2
Dear reviewer, thanks for the comments provided and for the time dedicated to reviewing our manuscript.
Below, we have provided a detailed response to each comment.
Thanks again.
- ABSTRACT:
- “Thus, this study aimed to understand the differences between the two standing strategies from a postural and neuromuscular perspective” - it will be better to put here the purpose of the work, which is included in the introduction section: to compare the bipedal position and the hand standing position by evaluating the two tasks from a stabilometric and muscular point of view in gymnasts.
- Please expand the abbreviations used in the abstract section (EMG, MDF)
Thank you for your comments. We have fixed the abbreviations and abstract.
- MATERIAL AND METHODS
- “Participants were included if they were free of injuries during the assessment period” - injuries they suffered earlier - e.g., a month before the study could have affected the results. Has this been considered? If not, it should be in the limitation section.
Thank you for your comment. Yes, we considered no injuries as inclusion criteria. We have also added such aspect within the limitations. Thanks
- “The study was approved by the local Bioethics Committee” - please provide the name of the organization issuing the approval (what university, etc.)
Thank you for your comment. We have added the University to the local Bioethics Committee
- “The surface electrodes, according to manufacturer instructions, were placed at a distance of 1 cm between them on the bellies of the gastrocnemius and the flexor carpi radialis muscles” - Was any electrode placement system used? (e.g., SENIAM standards)
Dear reviewer, the positioning of the electrodes was performed according to the instructions provided by the electrode manufacturer and by Atlas of muscle innervation zones, Barbero et al., 2012. The skin was shaved, a 70% alcohol solution was used to remove skin impurities. Each electrode, for each participant was positioned only once.
- Statistical analysis - please provide a program for statistical analysis, was the sample size calculated? Describe the tests used in detail.
Dear reviewer, we apologize for this issue. We added the required information. Thanks for the comment.
Reviewer 3 Report
Thank you for the opportunity to review I have the following comments on the text.
Throughout the text, the font should be standardized the abstract is written in two different fonts (L20 -23 and L 24-35) , the main text also (for example L167).
L20 – ‘’ Bipedal upright postural stance is not the only form of human standing’’ - Bipedal position is the only physiological position. The beginning of the sentence, in my opinion, misleads readers.
L24 – ‘’ skilled’’ - What do the authors think it means? The word skilled are a very subjective term.
L85 - Please add a research hypothesis.
L88 – ‘’skilled’’ – ‘’ skilled’’ - What do the authors think it means? The word skilled are a very subjective term.
L88 –‘’13 ‘’- Why did the authors decide on this number of participants? Please add sample size calculations.
L88 – ‘’20.2 ± 4’’ - Why did the authors decide on this age range ?
L91 – ‘’ three years in training’’ - There is not enough information here, you should add information about workouts on a weekly basis (how many workouts and total time in hours).
L100 – ‘’ maximal voluntary contraction (MVC)’’ - Please provide information on how the MCV was calculated and the appropriate citation for this.
L109 – ‘’ light warm-up’’ - Describe in detail.
L103 - EMG evaluation
· How much percent alcohol was used to cleanse the skin?
· Placement of electrodes according to the manufacturer's instructions is not an international guideline for electromyographic testing. Electrodes should be placed according to the SENIAM program (Surface ElectroMyoGraphy for the Non-Invasive Assessment of Muscles).
· The description of the exact placement of the electrodes is missing.
· After describing the study, I conclude that it is not EMG but sEMG. EMG study suggests using needles to test muscle acitvity. Superficial electrodes are used in surface electromyography ‘’sEMG’’. Superficial electrodes are used in surface electromyography sEMG
· L134 – ‘’surface electrodes’’ - accurately provide information about the electrodes. Conductive surface and manufacture.
· L132 –‘’ OTBio Quattro EMG device’’ - Does this equipment use a reference electrode ? if so add a description of the position of this electrode.
Statistical analysis
· Please add, effect size and confidence interval.
10.4300/JGME-D-12-00156.1
· L148 - Please describe what normality came out.
· What program was used for the analysis?
· With such a small group, shouldn't a non-parametric test be used?
Results
· ‘’Ns’’ - please enter full value of ''p''.
Conclusion - Should be rewritten in its entirety. They represent too much repetition of results.
References - Incorrect or incorrect notation of references: 8 , 12 , 25, 33 and 34.
Author Response
REVIEW 3
Dear reviewer, thanks for the comments provided and for the time dedicated to reviewing our manuscript.
Below, we have provided a detailed response to each comment.
Thanks again.
- Throughout the text, the font should be standardized the abstract is written in two different fonts (L20 -23 and L 24-35) , the main text also (for example L167).
Thank you for your comment, we have updated the font.
- L20 – ‘’ Bipedal upright postural stance is not the only form of human standing’’ - Bipedal position is the only physiological position. The beginning of the sentence, in my opinion, misleads readers.
Thanks for your comment. We've made the initial sentence clearer.
- L24 – ‘’ skilled’’ - What do the authors think it means? The word skilled are a very subjective term.
Thanks for your important comment. We have changed the term with a more specific one.
- L85 - Please add a research hypothesis.
Thanks for your important comment. We have added the research hypotheses as required.
- L88 – ‘’skilled’’ – ‘’ skilled’’ - What do the authors think it means? The word skilled are a very subjective term.
Thanks for your important comment. We have changed a term with a more specific term.
- L88 –‘’13 ‘’- Why did the authors decide on this number of participants? Please add sample size calculations.
Dear reviewer, unfortunately, we did not decide to use such a sample size, rather these were the only gymnasts available that we were able to recruit. Such aspect has already been acknowledged within the limitation section.
- L88 – ‘’20.2 ± 4’’ - Why did the authors decide on this age range ?
Dear reviewer, as for the previous comment, we were able to identify a specific cohort of gymnasts and we performed the experiments on those available.
- L91 – ‘’ three years in training’’ - There is not enough information here, you should add information about workouts on a weekly basis (how many workouts and total time in hours).
Dear reviewer, all of the included gymnasts had at least 3 years of training. These regularly exercised on a daily basis from Monday to Friday with each workout of approximately 1 hour.
- L100 – ‘’ maximal voluntary contraction (MVC)’’ - Please provide information on how the MCV was calculated and the appropriate citation for this.
Dear reviewer a brief paragraph explaining the procedure of the MVC was included. Thanks for the comment.
- L109 – ‘’ light warm-up’’ - Describe in detail.
Dear reviewer the warmup consisted of 1 set of stretching of the targeted muscles of 30 seconds duration. Such information is reported in the study design section. Such choice was determined since static stretching of longer durations is known to reduce performance and neural drive (doi.org/10.1242/jeb.229922)
- L103 - EMG evaluation
Dear reviewer, the section has been revised accordingly. Thanks for the comment.
- How much percent alcohol was used to cleanse the skin?
Dear Reviewer, A 70% alcohol solution was used to remove skin impurities.
- Placement of electrodes according to the manufacturer's instructions is not an international guideline for electromyographic testing. Electrodes should be placed according to the SENIAM program (Surface ElectroMyoGraphy for the Non-Invasive Assessment of Muscles).
Dear Reviewer, the recommendations provided by Atlas of muscle innervation zones, Barbero et al., 2012 for electrode placement have been followed. Each electrode, for each participant, was placed only once.
- The description of the exact placement of the electrodes is missing.
Dear reviewer, as requested we have explained electrode placement. Thanks for your comment.
- After describing the study, I conclude that it is not EMG but sEMG. EMG study suggests using needles to test muscle acitvity. Superficial electrodes are used in surface electromyography ‘’sEMG’’. Superficial electrodes are used in surface electromyography sEMG
Thank you for your important comment. We have modified EMG in sEMG throughout the manuscript.
- L134 – ‘’surface electrodes’’ - accurately provide information about the electrodes. Conductive surface and manufacture.
Dear reviewer, the section concerning sEMG evaluation has been updated. We hope that such amendments provide useful information. Thanks.
- L132 –‘’ OTBio Quattro EMG device’’ - Does this equipment use a reference electrode ? if so add a description of the position of this electrode.
Dear reviewer, the section concerning sEMG evaluation has been updated. We hope that such amendments provide usefull information. Thanks.
Statistical analysis
- Please add, effect size and confidence interval.
Dear reviewer, effect sixe and confidence intervals have been added as suggested. Thanks.
- L148 - Please describe what normality came out.
Dear reviewer. Thank you for the comment. A brief sentence was added to the method section.
- What program was used for the analysis?
Dear reviewer, we apologize for this issue. We added the required information. Thanks for the comment.
- With such a small group, shouldn't a non-parametric test be used?
We used parametric assessment according to the results of the normality tests. Thanks for the comment.
- Results
- ‘’Ns’’ - please enter full value of ''p''.
Dear reviewer, the tables have been edited when appropriate. Thanks.
- Conclusion - Should be rewritten in its entirety. They represent too much repetition of results.
Thank you for your comment. We have modified the conclusion.
- References - Incorrect or incorrect notation of references: 8 , 12 , 25, 33 and 34.
Thank you for your comment. The references have been updated.
Round 2
Reviewer 1 Report
This is a revised version of the manuscript that I reviewed earlier. The authors have improved the presentation of their work and provided important information to better assess the experiment and associated results. I still have some comments below that require attention before this manuscript can be recommended from my side for publication.
Line 191: Were there any non-parametric tests conducted? I have not read anything about the normality of the data when going through the results. If there is any non-parametric test, did the authors indeed calculate Cohen’s d for effect size? If yes, this is not a reliable effect size measure (e.g., https://journals.plos.org/plosone/article?id=10.1371/journal.pone.0239623)
Line 212: Please make clear that you do not refer to the rear foot but to the rear part of the foot (same for the front part of the foot/hand). When referring to the rear hand, the authors may also want to use the more colloquial term ‘palm’ so that the reader is not confused that one hand/foot was further to the back relative to the other. Related to this topic, how exactly do the authors define the border between front and rear parts of the hand/foot? This should be explained in the manuscript.
Figures 3 and 4: The captions are not complete. What RMS is calculated, of which measure? And what is the percentage of RMS that is mentioned in the caption of Figure 3? These need to be more explicitly described in both captions. Also, what do the error bars represent? This needs to be added in the caption.
Figure 5 is odd. Does the lower panel mean that during handstanding there is no EMG signal? What exactly do the panels show? EMG from the leg during upright and hand stance? EMG from the hand during upright and hand stance? I recommend having a figure with two panels showing EMG from the leg during the two stances and another two panels with EMG from the forearm during the two stances.
In addition, the authors need to report whether this is an individual subject or some averaging has taken place. If the former, this should be stated. Ideally the authors would present average traces across subjects with shaded areas around indicated variability.
The figure is also quite pixelated and the quality should be improved. Ideally, the label A and B should be on the upper left part of each panel.
Figures: I generally recommend adding the individual data point along the bar graphs so that data transparency is improved and the distribution of the data becomes evident.
It is still unclear what does the analysis in the frequency domain can tell us. A motivation for conducting this analysis, as well as a discussion of the no systematic effects on this variable are required.
Line 253: How can a larger CoP area be caused by smaller supporting surface? This should be better explained.
Minor:
Line 149: MVC is not defined yet, please first define all abbreviations before using them. Same for line 281 (MDF).
Line 244: ‘differs’ should be ‘differ’. Please proofread for further typos (e.g., line 312/participants, line 314/athletes).
Author Response
This is a revised version of the manuscript that I reviewed earlier. The authors have improved the presentation of their work and provided important information to better assess the experiment and associated results. I still have some comments below that require attention before this manuscript can be recommended from my side for publication.
Dear reviewer, thanks for providing useful comments which have improved the scientific quality of our manuscript. Below we have provided a detailed response to each of the additional comments. Thanks again
Line 191: Were there any non-parametric tests conducted? I have not read anything about the normality of the data when going through the results. If there is any non-parametric test, did the authors indeed calculate Cohen’s d for effect size? If yes, this is not a reliable effect size measure (e.g., https://journals.plos.org/plosone/article?id=10.1371/journal.pone.0239623).
Dear reviewer, as indicated within the method section, both parametric and non-parametric data were conducted when appropriate according to the previously performed normality test. These have been indicated within tables for clarity. Further, the reported effects size are Cohen’s d. However, in light of the reviewer’s comment we have modified the reported values for the non-parametric data with biserial rank correlations effect sizes. The method section has been also updated accordingly.
Thanks.
Line 212: Please make clear that you do not refer to the rear foot but to the rear part of the foot (same for the front part of the foot/hand). When referring to the rear hand, the authors may also want to use the more colloquial term ‘palm’ so that the reader is not confused that one hand/foot was further to the back relative to the other. Related to this topic, how exactly do the authors define the border between front and rear parts of the hand/foot? This should be explained in the manuscript.
Dear reviewer, thank you for your comment. We have adjusted the terms. Further, the determination of the pressure points of the rear/front aspect of the foot or hand within the baropodometric evaluation is an automatized feature of the platform/software (FreeStep v.1.0.3 software). The software determines the anterior or posterior aspects based on the acquired image and divides this into two equal 50% parts (50% rear and 50% anterior). We have added a sentence in the methods section to clarify such aspect. Thanks again.
Figures 3 and 4: The captions are not complete. What RMS is calculated, of which measure? And what is the percentage of RMS that is mentioned in the caption of Figure 3? These need to be more explicitly described in both captions. Also, what do the error bars represent? This needs to be added in the caption.
Figures captions have been modified as suggested. Thanks
Figure 5 is odd. Does the lower panel mean that during handstanding there is no EMG signal? What exactly do the panels show? EMG from the leg during upright and hand stance? EMG from the hand during upright and hand stance? I recommend having a figure with two panels showing EMG from the leg during the two stances and another two panels with EMG from the forearm during the two stances.
In addition, the authors need to report whether this is an individual subject or some averaging has taken place. If the former, this should be stated. Ideally the authors would present average traces across subjects with shaded areas around indicated variability.
The figure is also quite pixelated and the quality should be improved. Ideally, the label A and B should be on the upper left part of each panel.
Dear reviewer,
The figure was a representation of a single subject of the amplitude during the handstand (A) and the amplitude during standing (B). However, after having a second thought regarding such an image, this does not really add valuable information to the manuscript results. Therefore, we have decided to remove it from the manuscript.
Figures: I generally recommend adding the individual data point along the bar graphs so that data transparency is improved and the distribution of the data becomes evident.
Dear reviewer,
We have approached modifying figures 3 and 4 according to the reviewer’s suggestion. However, figure 3 with individual points becomes incomprehensible since for the H all individual values fall within a very small range and the bar becomes indistinguishable. We have included such a figure, although we believe the previous version was much clearer. Thanks.
It is still unclear what does the analysis in the frequency domain can tell us. A motivation for conducting this analysis, as well as a discussion of the no systematic effects on this variable are required.
Dear reviewer, we believe the frequency domain can provide information regarding fatigue and MU recruitment. The two tasks differ in nature, and HS is a complex task which requires both technical skill and important neuromuscular adaptations to be correctly performed. Our hypothesis was that despite the small window of analysis differences could have emerged. This was not observed in the present investigation. Two main aspects can be considered. The first is that 5 seconds are not enough to observe decreased MDF of the involved muscles, despite the high amplitude observed. The second is that the gymnast analyzed are used to hand standing and together with the short window we were not able to detect differences among the lower and upper limbs. We believe this aspect needs to be included in order to provide future experimenters with useful information for study designs in which sEMG is used in gymnasts. We have discussed such aspects within the discussion section. Thanks
Line 253: How can a larger CoP area be caused by smaller supporting surface? This should be better explained.
Dear reviewer, the sentence has been modified. Thanks
Minor:
Line 149: MVC is not defined yet, please first define all abbreviations before using them. Same for line 281 (MDF).
Dear reviewer, MVC had been previously defined within the study design section, as well as the MDF which was previously defined in the sEMG evaluation section.
Thanks
Line 244: ‘differs’ should be ‘differ’. Please proofread for further typos (e.g., line 312/participants, line 314/athletes).
Dear reviewer, thank you for your comment. We have adjusted the terms.
Thanks again for your comments and suggestions.
Reviewer 2 Report
Dear Authors,
Congratulations on making a significant change in the presented work.
In its current form, the work meets the criteria for publication in the Journal of Functional Morphology and Kinesiology.
Kind regards
Author Response
Dear Authors,
Congratulations on making a significant change in the presented work.
In its current form, the work meets the criteria for publication in the Journal of Functional Morphology and Kinesiology.
Kind regards
Dear reviewer. Thank you for helping us improve the quality of the manuscript.
Reviewer 3 Report
Thank you for the authors response. I am impressed with the improvements especially in the results section. After reviewing the paper again, I have the following comments.
L22 – ‘’experienced’’ - sounds better, however, suggests specifications – ‘’13 gymnasts with three years of sports experience’’. Note applies to the entire text.
L23 – ‘’ electromyographic ‘’ - should be changed to ‘’surface electromyography’’. The very word ''electromyography'' is not a device specification. Note applies to the entire text.
L41 - Suggests combining paragraphs. One sentence as a separate paragraph looks bad.
L83 – ‘’ electromyographic ‘’ - should be changed to ‘’surface electromyography’’. The very word ''electromyography'' is not a device specification. Note applies to the entire text.
L95 – Add clear information about the gender of the respondents.
L99 - it is acceptable.
L204 - suggests separating the two graphics, be placed side by side.
L224 - One signature should be under the graphic - combine the two signatures.
L331- 316 – limitations:
‘’ Dear reviewer, unfortunately, we did not decide to use such a sample size, rather these were the only gymnasts available that we were able to recruit. Such aspect has already been acknowledged within the limitation section.’’
Dear authors, I understand the difficulties in recruiting a study group. This is a common problem among researchers. However, it cannot be an argument explaining the small number of people in the study. Every researcher faces this. I would suggest expanding the study to other research units that may have access to new gymnasts.
If you are not able to meet this criterion in addition to the limitation suggests adding in the title the information - pilot studies or A Preliminary Report.
‘’Dear reviewer, as for the previous comment, we were able to identify a specific cohort of gymnasts and we performed the experiments on those available.’’
Dear authors, this is still not an explanation supported by science. You had access to study this group and examined them, but from your explanation it seems that age and gender were random. You simply had access to them. There is no scientific support why exactly you studied them (age and gender) ?
For example - more common injuries to ths grup of this age doing gymnastics ?
Conclusions still need to be improved.
References - still not corrected. 8, 17, 29 - pages are missing.
Author Response
Thank you for the authors response. I am impressed with the improvements especially in the results section. After reviewing the paper again, I have the following comments.
Thank you for taking the time to review our manuscript.
L22 – ‘’experienced’’ - sounds better, however, suggests specifications – ‘’13 gymnasts with three years of sports experience’’. Note applies to the entire text.
Dear reviewer, thank you for your comment. We have changed the reference in the text.
L23 – ‘’ electromyographic ‘’ - should be changed to ‘’surface electromyography’’. The very word ''electromyography'' is not a device specification. Note applies to the entire text.
Dear reviewer, thank you for your comment. We have changed the reference in the text.
L41 - Suggests combining paragraphs. One sentence as a separate paragraph looks bad.
Dear reviewer, thank you for your comment. We have combined the 2 paragraphs. Thanks
L83 – ‘’ electromyographic ‘’ - should be changed to ‘’surface electromyography’’. The very word ''electromyography'' is not a device specification. Note applies to the entire text.
Dear reviewer, thank you for your comment. We have changed the reference in the text.
L95 – Add clear information about the gender of the respondents.
Dear reviewer, thank you for your comment. We have added the gender information of the participants
L204 - suggests separating the two graphics, be placed side by side.
Dear reviewer, thank you for your comment. We believe that putting the two graphics side by side does not make it simple to identify that the reference points used for the two tasks are the same. With this representation, in our opinion, it is much clearer both the experimental setting and the pressure point distribution differences between the two tasks. Thanks again.
L224 - One signature should be under the graphic - combine the two signatures.
Dear reviewer, figure 3 and figure 4 are two separate images. At this point, we are unable to determine if the publisher will leave the two images as shown in the present version of the manuscript or they will be placed differently. Therefore, we would prefer to leave the two captions separate. Thanks
L331- 316 – limitations:
‘’ Dear reviewer, unfortunately, we did not decide to use such a sample size, rather these were the only gymnasts available that we were able to recruit. Such aspect has already been acknowledged within the limitation section.’’
Dear authors, I understand the difficulties in recruiting a study group. This is a common problem among researchers. However, it cannot be an argument explaining the small number of people in the study. Every researcher faces this. I would suggest expanding the study to other research units that may have access to new gymnasts. If you are not able to meet this criterion in addition to the limitation suggests adding in the title the information - pilot studies or A Preliminary Report.
Dear reviewer, unfortunately at this point it is not possible for us to continue sampling, therefore we accept your suggestion and have included a pilot study within the study title. Thanks again for your suggestion.
‘’Dear reviewer, as for the previous comment, we were able to identify a specific cohort of gymnasts and we performed the experiments on those available.’’ Dear authors, this is still not an explanation supported by science. You had access to study this group and examined them, but from your explanation it seems that age and gender were random. You simply had access to them. There is no scientific support why exactly you studied them (age and gender) ? For example - more common injuries to ths group of this age doing gymnastics?
Dear reviewer, thank you for your comment. Our study was not intended to target injuries or other characteristics which may reflect development (i.e. children). Our aim was to investigate if any differences were present between postural and neuromuscular characteristics during handstands and common standing. To this extent, gymnasts are the sporting individuals who better reflex such sample, being almost the only ones who are able to properly handstand. Within this context, we recruited young healthy gymnasts who at the time of the evaluations were free of injuries. We believe that if other populations (other sporting athletes, children or older adults, subjects with injuries or pathologies) were recruited, the results would not reflect typical standing vs handstanding strategies. Thanks again
Conclusions still need to be improved.
Dear review, thank you for your comment. We have modified the conclusion.
References - still not corrected. 8, 17, 29 - pages are missing.
Dear reviewer, we were not able to identify such information from the publishers. We apologize.
Thanks again for taking some time to review our manuscript.
Round 3
Reviewer 3 Report
Accepts authors' answers and manuscript.